# Novel Insights in Anti-CD38 Therapy Based on CD38-Receptor Expression and Function: The Multiple Myeloma Model

**DOI:** 10.3390/cells9122666

**Published:** 2020-12-11

**Authors:** Beatrice Anna Zannetti, Angelo Corso Faini, Evita Massari, Massimo Geuna, Enrico Maffini, Giovanni Poletti, Claudio Cerchione, Giovanni Martinelli, Fabio Malavasi, Francesco Lanza

**Affiliations:** 1Hematology Unit and Romagna Transplant Network, Ravenna Hospital, 48121 Ravenna, Italy; beatrice.zannetti2@libero.it (B.A.Z.); enrico.maffini@auslromagna.it (E.M.); 2Department of Medical Science, University of Torino and Fondazione Ricerca Molinette, 10129 Torino, Italy; angelo.faini@edu.unito.it (A.C.F.); fabio.malavasi@unito.it (F.M.); 3Flow Cytometry Unit, Laboratory Service, 47522 Pievesestina, Italy; evita.massari@auslromagna.it (E.M.); giovanni.poletti@auslromagna.it (G.P.); 4Laboratorio di Immunopatologia, A.O. Ordine Mauriziano, 10129 Torino, Italy; massimo.geuna@gmail.com; 5IRST/IRCCS, Cancer Center, 47014 Meldola, Italy; claudio.cerchione@irst.emr.it (C.C.); giovanni.martinelli@irst.emr.it (G.M.)

**Keywords:** CD38, plasmacells, multiple myeloma, CD38 antigen expression in various tissues, bone marrow microenvironment, anti-CD38 monoclonal antibodies

## Abstract

Multiple myeloma (MM) is a hematological disease characterized by the proliferation and accumulation of malignant plasmacells (PCs) in the bone marrow (BM). Despite widespread use of high-dose chemotherapy in combination with autologous stem cell transplantation (ASCT) and the introduction of novel agents (immunomodulatory drugs, IMiDs, and proteasome inhibitors, PIs), the prognosis of MM patients is still poor. CD38 is a multifunctional cell-surface glycoprotein with receptor and ectoenzymatic activities. The very high and homogeneous expression of CD38 on myeloma PCs makes it an attractive target for novel therapeutic strategies. Several anti-CD38 monoclonal antibodies have been, or are being, developed for the treatment of MM, including daratumumab and isatuximab. Here we provide an in-depth look at CD38 biology, the role of CD38 in MM progression and its complex interactions with the BM microenvironment, the importance of anti-CD38 monoclonal antibodies, and the main mechanisms of antibody resistance. We then review a number of multiparametric flow cytometry techniques exploiting CD38 antigen expression on PCs to diagnose and monitor the response to treatment in MM patients.

## 1. Introduction

Identification of cell-surface molecular targets in normal and neoplastic plasmacells was the result of a long-lasting quest by basic scientists from apparently disparate areas of research and which ultimately led to the development of antibody-based therapies. Antibody-mediated therapy based on CD38 as the molecular target has found wide application in clinics, with positive outcomes and good acceptance among patients. These successes tend to overshadow the fact that only a limited number of the molecule’s structural and functional features have been exploited in drug design. This work revisits the biological background and the research literature on CD38 and focuses on those aspects with potential for translating to clinical applications or for enhancing clinical outcomes. We believe that flow cytometry techniques may be useful in this regard.

## 2. Historical Background

In the 1970s, the characterization of surface molecules to be used for identifying human plasmacells (PCs) was one of the main objectives of the CD Workshop on International Workshop on Human Leucocyte Differentiation Antigens using murine monoclonal antibodies (mAbs) as probes [1]. CD38 was identified during attempts to define the structure and function of the T cell receptor (TCR) [2]. Early findings indicated that CD38, a transmembrane 46 kDa type II glycoprotein, was dominantly expressed by thymocytes, by activated T lymphocytes, and by some acute leukemias [3]. The initial designation as a “T cell activation” molecule had to be revised after thorough analysis of its tissue distribution—the CD38 an tigen is almost ubiquitous but is expressed at significantly different levels among tissues. Of the mononuclear cells in peripheral blood (PB) and bone marrow (BM), plasmablasts and PCs (as well as their neoplastic counterparts) display the highest cell surface density [4].

Uncovering the function(s) exerted by the molecule proved to be a complex task. One strategy was to trace its distribution in phylogeny and ontogeny and to observe its distribution in selected diseases, predominantly leukemias. Another approach was to study animal models lacking CD38 (known as CD38-knockouts, or CD38 KO). The study on CD38 KO mice indicated the effects attributable to the absence of the molecule. KO mice displayed impaired immune response in respiratory districts, and other problems. However, genetic ablation of CD38 in mice was noted to be insufficient for interfering with relatively-normal development, reproduction, and parental care [5]. These studies widened the scientific approaches to CD38 and diversified its applications. The most striking results came from neurophysiology, where CD38 emerged as a regulator of oxytocin release [6,7].

Experiments with murine anti-CD38 mAbs with agonistic properties (i.e., capable of delivering positive signals upon binding) were more telling. T lymphocytes were seen to be activated after exposure to agonistic anti-CD38 mAb [8]. Only a small fraction of anti-CD38 mAbs transduced signals upon binding, from which it followed that CD38 is a receptor. This led to the search for the counter receptor or ligand, which was later identified as CD31, a molecule known as platelet endothelial cell adhesion molecule-1 (PECAM-1) [9].

A breakthrough was then made by H.C. Lee, who identified a soluble enzyme purified from the mollusk *Aplysia californica* that was able to metabolize nicotinamide adenosine dinucleotide (NAD^+^) to produce adenosine diphosphate ribose (ADPR) and cycling ADPR (cADPR) [10]. The sequencing of this enzyme led to the discovery of a surprising similarity between the cytoplasmic enzyme from *Aplysia* and the human cell surface molecule CD38, in spite of a phylogenetic distance of approximately 950 million years [11].CD38 overall topology is similar to the related protein *Aplysia* ADP-ribosylcyclase. However, a disulphide bond (Cys119–Cys201) is unique in CD38 and helps stabilize its structure together with five other pairs of disulphide bonds (Cys67–Cys82, Cys99–Cys180, Cys160–Cys173, Cys254–Cys275, and Cys287–Cys296) conserved in the other members of the ADP-ribosylcyclase family [12].

Both murine and human CD38 were confirmed as the enzymes with the expected specificities [13]. Even if considered highly unusual molecules, ectoenzymes are now known to make up almost 5% of all cell surface molecules.

CD38 was analyzed in different physiological conditions, in pathology and in different species. Analysis of different human leukemias also gave fruitful results, especially in B-cell chronic lymphocytic leukemia (B-CLL).CD38 is a dependable marker for a subset of B-CLL patients and provides differential prognostic information [14]. The study of the signals implemented by CD38 led to the conclusion that the molecule acts as a molecular bridge to the microenvironment, supporting survival and proliferation over apoptosis [15]. Indeed, an acidic tumor microenvironment (pH ~ 5.5) significantly favors the CD38/CD203a/CD73 pathway, which leads to the production of adenosine (ADO). Instead, the canonical pathway is almost inefficient at this pH [16].

## 3. Monoclonal Antibodies in Multiple Myeloma Clinics

CD38 has found widest application in multiple myeloma (MM), where the particularly high surface density of the molecule makes it well-suited for antibody targeting. The following section will therefore examine the multiple roles played by CD38 in MM, particularly those with a potential impact on improving therapeutic applications.

Multiple myeloma, the second-most-common hematological disease, is a clonal B-cell disorder of PCs, with accumulation of transformed cells in the BM. Malignant PCs induce lytic lesions in the bone tissue and produce high amounts of monoclonal immunoglobulins in the serum and urine. Evidence of end-organ damage comes in the form of acute renal failure, hypercalcemia, and anemia [17].

The high and constant expression of CD38 by the majority of MM patients warranted success in antibody therapy. Good results were obtained in vivo through the therapeutic use of daratumumab (a human anti-CD38 IgG1 from Janssen, the first mAb entering clinical trials) as a monotherapy. The antibody was quickly approved by the various American and European regulatory agencies [18]. Later, the antibody was adopted in combination with immunomodulatory drugs or proteasome inhibitors. Similar positive results were more recently obtained with isatuximab (a humanized IgG1mAb, Sanofi, Paris, France) [19]. Comparable results were obtained in MM patients treated with MOR202 in monotherapy (Morphosis) [20]. TAK-079, a human IgG1λ (Takeda, Tokyo, Japan) is in phase 1/2a trial. The high potency of this antibody likely reflects its selective binding to myeloma cells and reduced binding to platelets and erythrocytes [21]. Comparative analysis between daratumumab and isatuximab are reported in literature [22,23]. However, no direct comparison between daratumumab and isatuximab versus MOR202 and TAK-079 is yet available, the latter drugs still not being available publicly. However, their detailed characteristics are reported in literature [20,21].

Antibodies exert their action both through Fc (Fragment crystallizable) and complement receptors and directly. The main mechanisms exploited by mAbs are as follows:

ADCC: antibody-dependent cellular cytotoxicity. The Fc portion of the therapeutic IgG is captured by specific receptors expressed by effector cells. This step triggers their lytic machinery and is followed by death of the tumor target.

ADCP: antibody-dependent cellular phagocytosis. The trigger mechanism is similar to that of ADCC, while the effector cells are monocytes and macrophages. The final step is opsonization of the tumor cell. Opsonization of tumor antigens or of antigen/antibody complexes may acquire relevance in the generation of autoimmune or self-vaccination responses [24].

CDC: complement-dependent cytotoxicity. The complement is an ancient component of innate immunity. Its interaction with the IgG bound to the target tumor triggers the complement cascade. The C1q component leads to the formation of a membrane-attack complex, while C3b contributes to the phagocytosis of tumor cells by phagocytic cells.

Direct actions: these effects are secondary to interaction between the antibody and the target antigen. Other effects are secondary to interactions with the Fc receptors (FcRs) expressed by surrounding cells. Most of the signals lead to myeloma-programmed cell death. Isatuximab is the only mAb able to deliver a signal in the absence of IgG cross-linking [25].

Effects on the immune system: the CD38 molecule is detectable on leucocyte populations in the blood, some endowed with opposing functions. CD38 is also detectable on platelets and erythrocytes, at very low density but over vast cell surfaces. In vivo experience shows that daratumumab works not only on the tumor target, but also on the immune effectors. Immunosuppressive populations such as T regulatory lymphocytes (Tregs) and B regulatory lymphocytes (Bregs) and myeloid-derived suppressor cells are eliminated, while T cell populations are increased [26]. Similar effects were reported for isatuximab, which produces effects also on NK (Natural Killer) cells [22].

Other interactions can take place between the therapeutic antibody and PCs. Recent observations showed that CD38 binding by daratumumab can be followed by important modifications in the cytoskeleton and membrane of MM cells. The CD38 membrane domain/antibody complexes can then be released as microvesicles (MVs), which are also rich in CD73, CD39, PD-L1 (a component of the immune network), and inhibitory complement receptors (CD55 and CD59). MVs can contribute to generating a tolerogenic microenvironment in the MM BM niche. Distantly, MVs are internalized in distinct FcR^+^ cell populations (NK cells, lymphocytes, and myeloid cells). The strategy of analysis following daratumumab internalization and generation of MVs is summarized in Figure 1.

MVs can also be internalized by dendritic cells. Such MVs may carry tumor antigens and activate dendritic cells against MM cells. This could lead to an improved immunological response consisting of a sort of autovaccination effect triggered by antibodies (Figure 1).

Moreover, recent approaches involving the use of antibodies for MM treatment deserve mention. Single domain antibodies (nanobodies) targeting CD38 are heavy-chain variable regions with affinity for CD38. Such nanobodies show encouraging efficacy in mediating MM cell lysis, in modulating CD38 enzymatic activity, and even in diagnostics [27,28,29]. Such nanobodies can also be exploited for generating CD38-targeting CAR-T cells which have already proved promising for mediating MM-cell killing [30,31].

## 4. Antibody Resistance

The anti-CD38 therapeutic strategy leads to a significant improvement in the survival of MM patients. However, after variable lengths of time, most patients relapse and develop drug resistance. The proposed mechanisms underlying antibody resistance have been summarized in recent reviews [32,33,34]. Here we consider some issues that may help improve therapy. There are several mechanisms that lead to resistance, namely:

(1) CD38 cell surface density. The therapeutic antibodies also exert their action (dependent or not on Fc) on effectors, whose CD38 expression differs only quantitatively from MM cells. In fact, these mechanisms show a primary dependence on the density of the CD38 molecules expressed by the target tumor [23,35]. Reduced surface expression is also an issue in clinical flow cytometry; expression levels are reported to drop rapidly after antibody treatment in all the main leukocyte populations in the PB. This can be overcome by inducing de novo expression of the molecule using all-*trans* retinoid acid (ATRA), an approved drug [36,37]. The results obtained through the combination of ATRA and daratumumab are now undergoing validation by clinical trial (NCT02751255). Recent observations indicate that the JAK–STAT (Janus Kinase-Signal Transducer and Activator on Transcription) axis is involved in CD38 expression by myeloma cells. The JAK–STAT1 pathway mediates CD38 upregulation, while the JAK–STAT3 pathway produces CD38 downregulation. Inhibition of the JAK–STAT3 pathway by ruxolitinib (a STAT3 and STAT1 inhibitor) has been proposed as a novel therapeutic option for increasing CD38 expression and ADCC defense [38].

(2) IgGFcRs. The Fc domain of the therapeutic immunoglobulin may interact with specific receptors expressed by different cell populations. FcRs are part of a family of surface molecules differing in terms of affinity and ability to transduce signals [39]. Myeloma does not express FcRs. Some FcRs are genetically polymorphic. This issue was analyzed by testing the effects of FCRG3A and FCRG2B on daratumumab treatment of MM patients. The results obtained are negligible in terms of therapeutic efficacy; this ruled out the selection of patients on the basis of genotype [40]. However, the issue of FcRs has not yet been fully answered, especially in NK cells. Flow cytometrists reported that the NK population disappears following daratumumab therapy. This effect persists long after the beginning of therapy, wiping out the most potent anti-tumor effectors. The low-affinity CD16 FcR has a role in the activation of NK cells as shown in vitro after CD38 engagement by murine-specific mAbs [41]. The role of CD16 has been further confirmed in vitro by human mAb while studying the mechanism of action of isatuximab [35]. A possible explanation for the peculiar disappearance of NK cells may be found in a recent set of data indicating that CD38 ligation by daratumumab is followed by internalization and degradation of the target molecule. This leaves the NK population still activated but lacking CD38 surface expression. These CD38-negative cells maintain their ability to induce monocyte activation, increase phagocytosis, and enhance the expression of T cell co-stimulatory molecules [42]. The role of FcRs in the immune response during antibody-mediated therapy was taken into account in the development of a next-generation anti-CD38 mAb, namely SAR442085 (Sanofi). This mAb has a genetically-modified Fc region and shows a higher affinity for CD16 FcRIIIa than daratumumab. The antibody is in Phase 1 trial [43]. Other aspects pertaining to the role of the aspecific ligation of therapeutic IgG can be gleaned from the study of neonatal FcR (FcRn), which physiologically controls the homeostasis of IgG and albumin. Its role in the multiple interactions taking place in vivo during antibody therapy is under investigation using TAK-079 [44].

(3) Epitope specificity. The sequence recognized by the different anti-CD38 mAbs is of dual relevance—firstly, it may influence the enzymatic functions of the molecule, and secondly, it may make it possible to re-treat patients using an antibody different from the one showing resistance. Inhibited enzymatic functions might represent an added value in therapy.

(4) Enzymatic function. As an enzyme, CD38 uses not only NAD^+^ as a substrate, but also adenosine triphosphate (ATP) when it is found in a closed system at a low pH together with CD203a, a surface ectoenzyme with ectonucleotidepyrophosphatase/phosphodiesterase-1 activity, also known as Plasma Cell-1 [45]. CD73 is the final element of the enzymatic chain and leads to the production of ADO [46]. The CD38/CD203a/CD73 axis may bypass the canonical CD39/CD73 pathway for ADO production. The final product, ADO, is bound by specific adenosine receptors (ADORs) that are expressed by different immune cells and by other components of the BM niche. The production of ADO causes a blockade of the immune cells and an increase in Tregs, stromal mesenchymal, and dendritic cells, all of which contribute to tumor immune escape. What is known at the moment is that the mAb isatuximab reacts with specificity with the catalytic site of CD38, blocking the production of cADPR and ADPR. Whether this has a positive effect on the immune response is still to be determined. Indirect evidence is that BM plasma of patients with the worst clinical prognosis contains high levels of ADO. In addition, ADO levels have been demonstrated to correlate with disease progression. This may be taken as an indication that the BM niche of MM patients is a site of ectoenzymatic activities [47].

The second key question is related to antibody retreatment, i.e., whether a second anti-CD38 antibody can be employed in case of resistance to the first, using a drug specific for a different CD38 epitope. In vivo experience with daratumumab has shown that both primary and secondary resistance may occur during anti-CD38 antibody therapy. Primary resistance is attributed to CD38 density on target cells and it also may be secondary to the expression of inhibitory complement proteins, providing protection against CDC and ADCC [33]. Isatuximab also appears very sensitive to the molecular density of the target [23]. At the moment, the issue of re-treatment of MM patients with an anti-CD38 mAb recognizing a different epitope in order to bypass resistance to the first lacks evidence, also considering the anti-CD38 antibodies that are now included in the therapeutic armamentarium.

To conclude this section, the evidence reviewed here confirms the pleiotropic role played by CD38 in MM. The efficacy of anti-CD38 therapy in inhibiting the survival of MM cells and in reverting immune suppression and bone disease is witnessed in the results of routine practice and clinical trials. Although many questions remain, flow cytometry is currently the best candidate to study how to solve many of the issues raised by the use of therapeutic mAbs.

## 5. Flow Cytometry Detection of CD38 Antigen Expression in Myeloma Cells from Different Sources (PB, BM, Mobilized PB, and Leukapheresis Products)

Flow cytometry (FC) methods have developed rapidly in recent years and now allow the simultaneous detection of a large number of surface and intracellular antigens. The ability to analyze a very large number of events in a very short time is a major advantage of FC over morphological and immunohistochemical techniques.

Flow cytometry is used for the diagnosis of PC disorders, as it detects abnormal PCs according to their aberrant immunophenotype. The most-frequent markers used for this purpose are the cytoplasmic immunoglobulin light chains, CD19, CD56, and CD45, but many other markers of interest and a number of immunophenotyping strategies have been described in the literature. The most important immunophenotypic markers are CD38 and CD138, which are co-expressed by normal and pathological PCs, thus allowing their identification in the BM, PB, or other tissues by using FC.

Growing data has revealed the existence of important phenotypic differences between normal and clonal PCs.

Normal PCs can be easily identified in BM aspirates, despite their relatively-low frequency (0.1–1% of total nucleated BM cells) through a combination of CD38, CD138, and CD45. In fact, normal PCs express CD38 at very high level, greater than any other hematopoietic cell, and co-express CD138 [48]. The cells gated on CD38+^high^/CD138+ and back-gated on CD45+^int/dim^ and angular scatter represent putatively all of the BM PCs (Figure 2). Normal PCs are clearly polyclonal for kappa and lambda light chains and phenotypically heterogeneous (Figure 2), as demonstrated by differential expression of CD19 and CD56 [49,50]. Use of these two markers makes it possible to identify four different subpopulations of PCs in the BM of healthy subjects: the CD19+/CD56− “normal” PCs (accounting for a median of 60.3%), the CD19 − ve/CD56 + ve “aberrant” PCs (9.6%), the CD19 − ve/CD56 + ve double-negative PCs (29.9%), and the CD19+/CD56+ double-positive PCs (3%) (Figure 2) [49]. The exact physiological significance of these subsets of normal PCs is not completely elucidated; the most accepted explanation is that the four phenotypes are associated with distinct maturational compartments of PCs and their relative frequency inversely correlates with increasing maturity [48].

Recognizing this heterogeneity is of key importance in identifying neoplastic PCs, especially in the analysis of minimal residual disease (MRD). Myeloma PCs often display aberrant phenotypes. The most frequent (and informative) aberrant markers include CD19 (negative), CD56 (positive), CD45 and CD38 (under expressed), CD27 and CD81 (negative or dimly expressed), and CD117 (positive) [51,52]. Nevertheless, many markers are not aberrant on their own, but are useful in identifying an aberrant phenotype. CD19 and CD56 are good examples of this. Tumor PCs are CD19-negative in virtually every MM case but, as stated above, there is also a significant fraction of normal PCs (30–40%) that are CD19-negative. Similarly, CD56 has long been claimed as a myeloma-specific marker, although a subset of normal PCs (9–12%) expressing CD56 has been identified [51,52].

Since a PC is able to produce a single type of immunoglobulin (isotypic restriction), myeloma PCs can be enumerated by their homogeneous expression of either kappa or lambda light chains. The confirmation of the clonal nature of myeloma PCs using the clonal restriction of cytoplasmic kappa or lambda light chains remains the most robust and highly sensitive method for assessing the neoplastic nature of PCs, particularly for MRD testing [53].

A very small subset of CD19+, CD20 − ve, CD38+^bright^, and CD138+/ − ve cells, accounting for less than 1% of the whole B cell compartment, is usually detectable in PB [54]. These cells are called plasmablasts, represent the circulating PCs compartment, and are considered immature PCs. Plasmablasts express CD27 and CD45 at high fluorescence intensity, are CD56-negative, and may retain very low levels of surface immunoglobulins (Figure 3). This phenotype is highly specific and, together with polyclonal immunoglobulin light chain expression, facilitates the identification of neoplastic PCs in the PB and PB derivatives (i.e., mobilized PB and leukapheresis products) more than in the BM.

Very recently, Paiva and colleagues have demonstrated in a prospective clinical trial that next-generation flow (NGF) MRD negativity can be a sensitive and widely-applicable end-point criterion for evaluating treatment efficacy in MM patients [55]. The EuroFlow panel has been designed in order to increase the sensitivity (10^−5^–10^−6^) and the standardization of multicolor FC [56,57]. This panel includes two 8-color tubes (tube 1: CD138, CD27, CD38, CD56, CD45, CD19, CD117, and CD81 and tube 2: CD138, CD27, CD38, CD56, CD45, CD19, cyIgκ, and cyIgλ). This consensus panel requires the acquisition of as many cells as possible (as many as 10^7^ cells or more). The second step is the evaluation of the limit of quantitation (LOQ, calculated as 50 clonal PCs among all nucleated cells) and the limit of detection (LOD, calculated as 20 clonal PCs among all nucleated cells). This method allows discrimination between positive and negative samples.

Finally, treatment with anti-CD38 mAbs such as daratumumab or isatuximab can reduce or abrogate the antigen expression on MM cells. This can be a limit for the use of CD38 during MRD assessment. A multi-epitope CD38 antibody used in an advanced FC panel can solve this problem, since this antibody binds to multiple epitopes of the CD38 molecule not covered by daratumumab. It may be important to know that CD38 surface downregulation can be bypassed by the analysis of intracellular CD38 through the same protocol as that used for intracellular kappa and lambda immunoglobulin chain staining [57]. It is known that surface CD38 is flanked by a population of molecules expressed on the inner side of the cell membrane, and taking part to the dynamics of the functions of the molecule [58].

Flow cytometry applied to MM plays a crucial role not only for research purposes but also for guiding clinical practice.

The main applications of next-generation flow cytometry in MM are illustrated in Table 1.

## 6. Conclusions

Experience acquired with human CD38 in vitro found unprecedented support from the availability of a human disease (namely, CD38+ MM) treated with a specific human antibody. This approach has been received positively by clinicians because of the significantly-improved patient outcome from antibody treatment. For basic scientists, the disease is an ideal model for confirming evidence from in vitro experiments, mainly conducted on cell lines but lacking the complex background of a whole organism.

Pulling together these different conclusions, it now appears that CD38 is not just a marker of PCs and MM, but a pleiotropic molecule, whose functions have yet to be fully defined. CD38 is quite a unique ectoenzyme located on the cell membrane, but the breakdown of its substrate NAD^+^ leads to the production of second messengers operating in the cytoplasm. Secondly, ATP, the major source of energy and signals in humans, can become a substrate for CD38 in the presence of CD203a, which belongs to another ectoenzymatic network. The switch to the second metabolic pathway leads to the generation of ADO—this step is favored by pH acidification driven by the tumor as one of its multiple escape strategies. In addition to these hypotheses, other solid evidence supports the view that the metabolism may be useful as a therapeutic co-target in myeloma therapy. This is demonstrated by the recent observation that bone marrow stromal cells influence the myeloma growth through the process of intercellular mitochondrial transfer, which is made physically possible by tumor-derived tunneling nanotubes, where CD38 also plays a role [59].

CD38 expression has also emerged as an attractive field of investigation for further refinement of anti-CD38 therapy. Indeed, the application of FC to MM plays a crucial role not only for research purposes but also for guiding clinical practice. FC has the ability to distinguish between normal, reactive, and malignant PCs, to assess the risk of progression from MGUS/SMM to MM, to provide information about treatment response (MRD detection), and, finally, to identify new phenotypic markers to be used for target therapy.

Flow cytometry may become instrumental in furthering our understanding of the in vivo behavior of the antibody, an area of research that is still in its early stages of investigation. FC seems particularly well-poised to provide decisive contributions in studies of the therapeutic antibodies affinities in patients. A necessary word of caution is that both CD38 and FcRs display genetic polymorphisms. Further, antibodies with low affinity may actually perform even better in vivo since they only bind the target when it is at high cell surface density. At the same time, the CD38 molecule is left untouched when present on innocent bystander cells. Another field of extreme interest is the study in patients of the different interactions between antibodies and distinct FcRs, each of which has unique binding properties and a specific ability to transduce signals [39]. Of central interest in clinical practice will be the analysis of modifications induced in the Fc domain of the therapeutic antibody in order to alter the life span of the IgG and of the signals implemented [60]. In these cases, FC will be key to monitoring the effects differentially detectable on the tumor target as well as on the components of the immune response. Building on studies initially conducted on the anti-CD20 mAb rituximab, the next-generation anti-CD38 mAb isatuximab is characterized by a genetically-modified Fc domain, which displays high affinity for CD16 and CD32a FcRs. The reported results show an increase in ADCC potential as well as in the lytic power of NK cells [43].

In conclusion, the story of CD38 serves as a paradigm of how the basic sciences have identified a very complex molecule (receptor, ectoenzyme, immunomodulator, and modulator of cellular energy metabolism) that can be exploited for antibody target therapy. A thorough knowledge of the biology of CD38 is essential in order to understand anti-CD38 mAbs mechanisms of action and mechanisms of resistance.

## Figures and Tables

**Figure 1 cells-09-02666-f001:**
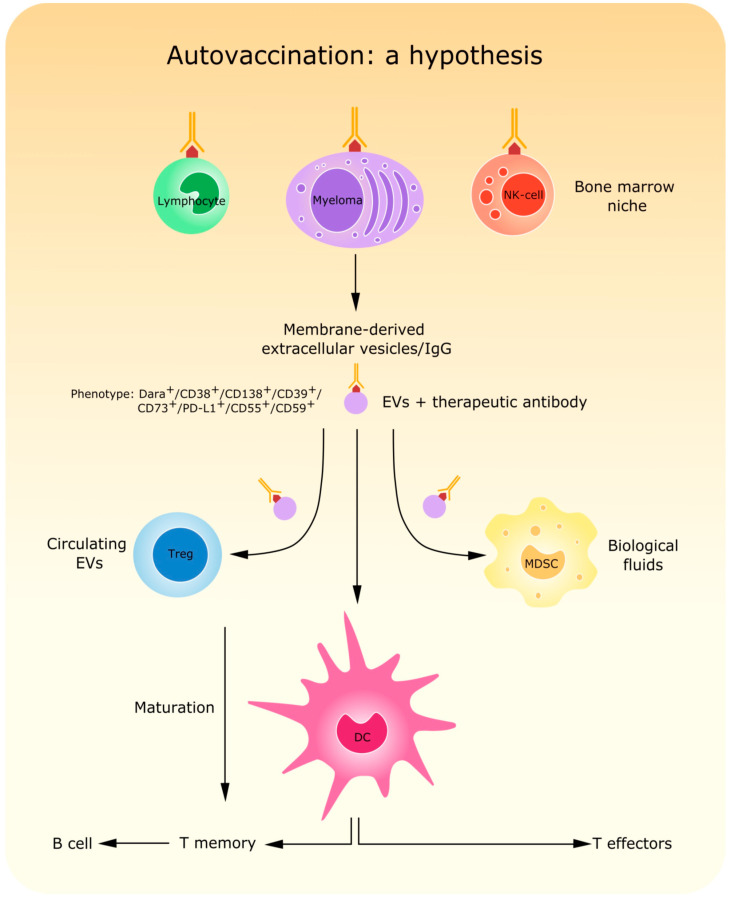
Schematic representation of the effects induced by therapeutic antibodies on multiple myeloma (MM) cells. The antibody-target ligation induces a redistribution of CD38 together with morphological modifications leading to release of antibody-covered microvesicles (MVs). Such MVs can be internalized by different subsets of immune cells, thus enhancing anti-tumor response and even potentially leading to a sort of autovaccination whose main actors are MV-activated dendritic cells.

**Figure 2 cells-09-02666-f002:**
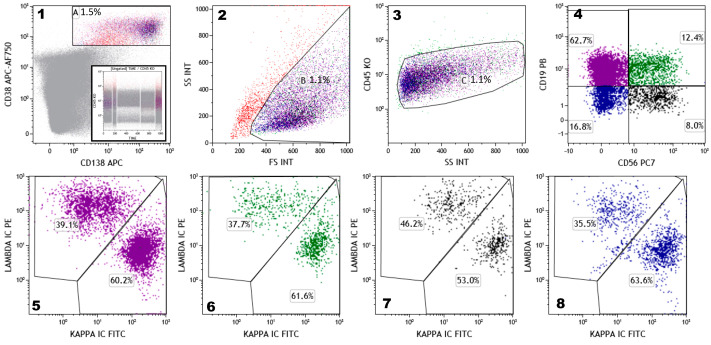
Normal bone marrow plasma cells (PCs) are identified by means of the co-expression of CD38^bright^ and CD138 (Plot **1**-gate A) backgated on forward vs. side scatter (Plot **2**-gate B) and then on side scatter and CD45^int^ (Plot **3**-gate C). The four populations of normal PCs (CD19+CD56−, violet; CD19+CD56+, green; CD19−CD56+, black; and CD19−CD56−, blue) and their relative proportions are identified in Plot **4** and for each of them the cytoplasmic kappa/lambda expression (polyclonal) is shown (Plots **5**–**8**). The analysis shown is the result of a data set merge of five different normal bone marrow samples (as depicted in the insert in Plot 1, where time vs. CD45 is shown).

**Figure 3 cells-09-02666-f003:**
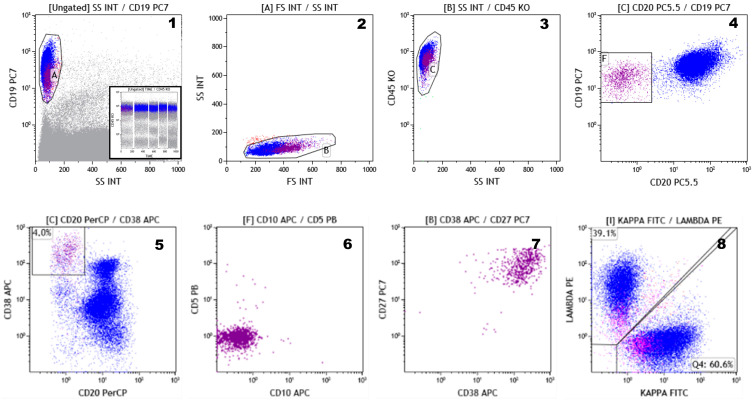
Peripheral blood plasmablasts analysis. The plasmablasts are identified as CD19+ve CD20 − ve cells (gating strategy is displayed in Plots **1**–**4**). This cell subset is CD20 − ve, CD38 + high, and represents only 4% (the average of five peripheral blood from healthy subject) of the entire CD19 + ve cells (Plot **5**), is negative for both CD5 and CD10 (Plot **6**), is strongly positive for CD38 and CD27 (Plot **7**), is expressed on the cell surface of both kappa/lambda immunoglobulin light chains (pink colored) at very low intensity when compared to the expression of kappa/lambda light chains on the other CD19+ cells (blue colored) (Plot **8**). The analysis shown is the result of a data set merge of five different normal peripheral blood samples (as depicted in the insert in Plot 1, where time vs. CD45 is shown).

**Table 1 cells-09-02666-t001:** Applications of next-generation flow cytometry in MM.

(1)	To identify and distinguish normal, reactive, and malignant PCs;
(2)	To monitor and assess the risk of progression from monoclonal gammopathy of unknown significance (MGUS)/smoldering multiple myeloma (SMM) to MM;
(3)	To guide and monitor hematopoietic stem cell collection in MM patients undergoing stem cell transplantation;
(4)	To provide information on minimal residual disease (MRD) status in different phases of the disease;
(5)	To identify new phenotypic markers to be used for myeloma target therapy;
(6)	To enumerate T-,B-, and NK-cell subsets, and myeloid-derived suppressor cells, in the peripheral blood (PB) of patients who have been treated with anti-CD38 therapy and/or bytes antibodies;
(7)	To assess the presence of tumor-associated macrophages (TAMs) in bone marrow (BM) samples in the various phases of the disease, thus providing insights on tumor microenvironment.

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
