# Peer review of "Novel Insights in Anti-CD38 Therapy Based on CD38-Receptor Expression and Function: The Multiple Myeloma Model"

_cells, 2020, doi:10.3390/cells9122666_

Round 1

Reviewer 1 Report

Figure 1 is difficult to read:

.....Letters are too small

.....The figure is "hazy"

Also the reference

(39 Atanackovic, D., et al. 2019) in the legend is not in the Lit.cited list !

Figure 1 should be improved

Author Response

Dear Reviewer 1,

Thank you for the positive evaluation of the paper by Zannetti et al.

1) The English of the manuscript has been improved by an English native speaker.

2) The presentation of the Results has been modified and improved.

3) Fig 1 has been improved and the letters enlarged and made less blurry.

4) Atanackovic is now included among the references.

Reviewer 2 Report

Major points.
1. The paper needs to revised. Each section should contain three to four paragraphs, each devoted to a particular subtopic. Redundant statements should be removed. The lengthy history of CD38 research is the most coherent section of the paper. However, this story has been presented also in other excellent reviews of the Malavasi lab. 

some specific suggestions:
i. Section 2: 
Focus on the role of CD38 in the MM. 
Add a § referring to Figure 1. 
Explain how an anti-inflammatory, acidic TME promotes tumor growth.
Move the description of CD38-specific Abs to section 3.

ii. Section 3: Begin with a description of the common and distinctive features of daratumumab, isatuximab, MOR202, TAK-079. Add a § on CD38-specific heavy chain antibodies (PMID : 27251573, 29084989, 32194826, 30524421) and a § on chimeric antigen receptors (PMID: 30185037, 28506593, 32194826, 31466926). 

iii. Section 4: This is largely redundant and superfluous.
The first three § would serve nicely as an introduction (could replace the lengthy history).
The § on microvesicles could be moved to section 2 and include a reference to Figure 1. 

iv. Section 5: Show and discuss data of MM samples. 

2. Figure 1:
The text in the figures is too small. 
The reference to this figure in the text is not intelligible, i.e. it needs to be rephrased. 

3. Figure 2: 
Data is from a single healthy individual. Add data from a larger number of individuals (at least n = 3), e.g. a bar diagram illustrating the % of plasma cells (panel A) and the ratio of plasma cell subsets (CD19+/- CD56+/-, panel 4) in several healthy individuals.
% are missing in panels 1 and 2. 
Show a corresponding, representative dataset for a BM sample from a MM patient.

4. Figure 3. 
Add a panel illustrating the expression of CD38 on plasmablasts vs. other B cells, i.e. CD20 on the X-axis, CD38 on the Y axis)
Data is from a single healthy individual. Add data from a larger number of individuals (at least n = 3), e.g. a bar diagram illustrating the % of plasmablasts (panel 4), % of CD38+ plasmablasts vs. B cells (new panel), and the ratio of plasma cell subsets (kappa/lambda, panel 6) in several healthy individuals.

% are missing in panels 1 and 4. 
Show a comparable analysis of a representative sample from a MM patient. 
Plasmablasts do not express polyclonal light chains (omit polyclonal in the legend).

5. Reference 30 is cited erroneously on multiple occasions. CD38 does not use ATP as substrate.

6. References 45 and 45 are cited erroneously: 
These papers show only data for a minor activity of CD38 (production of cADRP or cGDPR), not for the main activity of CD38 (production of ADPR).

7. Antibodies may still be attached to CD38 on cells obtained from Ab-treated patients. Flow cytometry with an antibody or nanobody that binds independently of the therapeutic antibody is better than use of a multiepitope polyclonal mix of antibodies (PMID 28522580)

Minor points

1. Consider a more appropriate title:  CD38-expression and utilization in the context of Multiple Myeloma and plasma cells

2. Introduction: 
Explain the conserved structure of the Aplysia cyclase and CD38 (same fold, five conserved disulfide bridges).  
Describe the phenotype of CD38ko mice.

3. The terms "demolition" "destruction" should not be used to describe enzyme activity. "hydrolysis" or "conversion to" is more appropriate.

4. Check the spelling and inappropriate use of capital lettering
plasmabasts, dartumumab, plots

5. Often, two words are not separated by a space. Similarly, a space is often missing after the period.

6. Use flow cytometry rather than "FC" to avoid confusion with Fc and PC.

Author Response

Please find below our point-by-point response to your review.

Major points:

Question 1

The paper needs to revised. Each section should contain three to four paragraphs, each devoted to a particular subtopic. Redundant statements should be removed. The lengthy history of CD38 research is the most coherent section of the paper. However, this story has been presented also in other excellent reviews of the Malavasi lab. 

Answer 1

We have revised the manuscript to make it more concise and made some revisions in term of the English issues according to your valuable suggestion. The English of the manuscript has been improved by an English native speaker. I would like to stress the fact that the manuscript originates in a common effort shared by basic and clinical scientists and intends to highlight synergizing evidence from distant fields, as requested by the editors of the Special Issue.

Some specific suggestions:

Question

  1. i) Section 2: Focus on the role of CD38 in the MM. Add a § referring to Figure 1. 
    Explain how an anti-inflammatory, acidic TME promotes tumor growth.
    Move the description of CD38-specific Abs to section 3.

Answer

  1. i) Section 2 has been revised according to your suggestions and specific references referring to Figure 1 have been added. The last paragraph of section 2 has been moved to section 3.

Question

  1. ii) Section 3: Begin with a description of the common and distinctive features of daratumumab, isatuximab, MOR202, TAK-079. Add a § on CD38-specific heavy chain antibodies (PMID : 27251573, 29084989, 32194826, 30524421) and a § on chimeric antigen receptors (PMID: 30185037, 28506593, 32194826, 31466926). 

Answer

  1. ii) Section 3 has been modified and a paragraph concerning nanobodies and CAR-T cells with proper references has been included. The same has been done while including the suggested comparison between anti-CD38 therapeutic antibodies. However, the data available especially concerning TAK079 and MOR202 and their comparison with other antibodies is limited. Such considerations have been added to the text as well as appropriated references.

Figure 1 has been moved at the end of section 3, being more appropriate here in our opinion. Indeed, the figure describes additional mechanisms of action of therapeutic antibodies and should be included in this section rather than in the “background” section. The text contained in the figure has been enlarged as well as the size of the figure itself.

Question

iii. Section 4: This is largely redundant and superfluous.
The first three § would serve nicely as an introduction (could replace the lengthy history).
The § on microvesicles could be moved to section 2 and include a reference to Figure 1. 
Answer

iii) Section 4 has been slightly modified. However, the part considered redundant has been substantially kept, as the description of resistance is one of the key point of the review and an excessive reduction of its content could have made the whole discussion not clear.

Question  

  1. Section 5: Show and discuss data of MM samples. 
    Answer

iv). Section 5. Thank you for the suggestion to show and discuss data of MM samples. We omit to show this data because a great number of previous paper and review on MM show the flow cytometric expression of different antigens and immunoglobulin light chains in myelomatous cells. We prefer to focus on the expression in normal plasma cells of those antigens that are largely employed to identify neoplastic plasma cells, highlighting that their expression on normal plasma cells requires to be extremely careful in the analysis of myeloma minimal residual disease. Section 5 has been omitted and its content redistributed in the text (see point 2) in order to make the reading smoother.

Two paragraphs have been added to clarify the points indicated as needing more explanation. Proper references have been included.
Question 2

Figure 1:
The text in the figures is too small. 
The reference to this figure in the text is not intelligible, i.e. it needs to be rephrased. 
Answer 2

Figure 1. Figure 1 has been improved and the letters enlarged and made less blurry.

Question 3

Figure 2: 
Data is from a single healthy individual. Add data from a larger number of individuals (at least n = 3), e.g. a bar diagram illustrating the % of plasma cells (panel A) and the ratio of plasma cell subsets (CD19+/- CD56+/-, panel 4) in several healthy individuals.
% are missing in panels 1 and 2. 
Show a corresponding, representative dataset for a BM sample from a MM patient.

Question 4.

Figure 3. 
Add a panel illustrating the expression of CD38 on plasmablasts vs. other B cells, i.e. CD20 on the X-axis, CD38 on the Y axis)
Data is from a single healthy individual. Add data from a larger number of individuals (at least n = 3), e.g. a bar diagram illustrating the % of plasmablasts (panel 4), % of CD38+ plasmablasts vs. B cells (new panel), and the ratio of plasma cell subsets (kappa/lambda, panel 6) in several healthy individuals.

Answer 3 and 4

Figure 2 and 3.

We appreciate the request to show data from a larger number of healthy individuals. Actually, the plots showed in this figure and in the figure 3 (from peripheral blood) are from 5 different healthy individuals. The single data set file, in which the 5 data sets are merged, is obtained from the function of Kaluza software “Merge selected Data Sets”, and allow the analysis of different files as a single one. Obviously, all the percentage obtained from this kind of analysis are the mean value of the different samples merged. 

We also modified the figures and the legends following the suggestion given.

We didn’t show a figure of a BM from MM patient for the same reason previously explained.

Question 5.

Reference 30 is cited erroneously on multiple occasions. CD38 does not use ATP as substrate.

Question 6.

References 45 and 45 are cited erroneously: 
These papers show only data for a minor activity of CD38 (production of cADRP or cGDPR), not for the main activity of CD38 (production of ADPR).

Answer 5 and 6.

The reference list has been revised.

Question 7.

Antibodies may still be attached to CD38 on cells obtained from Ab-treated patients. Flow cytometry with an antibody or nanobody that binds independently of the therapeutic antibody is better than use of a multiepitope polyclonal mix of antibodies (PMID 28522580)

Answer 7.

Antibodies. The chapter has been extensively modified.

Minor points:

Question 1.

Consider a more appropriate title:  CD38-expression and utilization in the context of Multiple Myeloma and plasma cells
Answer 1.

title. The title seems to us appropriate An alternative title may be:

EFFECTS OF CD38 RECEPTOR MEASUREMENT ON MULTIPLE MYELOMA THERAPY: MUTUAL SYNERGIES BETWEEN BASIC AND CLINICAL SCIENCE

Question 2.

Introduction: Explain the conserved structure of the Aplysia cyclase and CD38 (same fold, five conserved disulfide bridges).  Describe the phenotype of CD38ko mice.
Question 3.

The terms "demolition" "destruction" should not be used to describe enzyme activity. "hydrolysis" or "conversion to" is more appropriate.
Question 4.

Check the spelling and inappropriate use of capital lettering plasmabasts, dartumumab, plots
Question 5.

Often, two words are not separated by a space. Similarly, a space is often missing after the period.
Question 6.

Use flow cytometry rather than "FC" to avoid confusion with Fc and PC.

Answer 2.3.4.6. The manuscript has been rewritten, accordingly.

Reviewer 3 Report

-General Structure:

# This review is a very difficult read. Each section is a collection of observations/statements and the authors quickly switch from one idea to the next. There seems to be lot of information but this needs to be organized to convey one main idea per section. At the moment there is lack of flow and clarity. Find the main idea that you wish to convey in each section and summarize findings from the literature with a good structure and flow between the statements. The only exception is section 5 concerning flow cytometry which reads coherently and is well structured.

-Issues with references:

#Several sentences lack proper references. For example, page 2 the paragraph starts with “Study of function…” and “Another approach…”. Similarly, page 6 “It has also been…”. These paragraphs do not have references. There are similar instances throughout the review.

#Another problem is that the authors cite reviews and when one goes back to the review that article lacks a proper reference as well. As an example in page 3 “One of the first molecules…” cites reference 20. Reference 20 simply states this fact without showing data or citing a reference. Cite the article that describes the original work showing expression of these two molecules. Check references thoroughly to insure original work is cited.

#What is 39 in Fig 1? There is no reference relating to Atanackovic D in the list of references.

#The authors have self-cited more than 20 times, which is excessive. Please be careful when self-citing.

-Content

#Provide some explanation regarding CDC, ADCC.

#Page 4 “These contradictory findings…” the scorpion effect is not clear. Explain it clearly.

#The authors provide a rather extensive history of CD38. This could be shortened and the space can be utilized to provide summary of the ongoing trials with CD38 monoclonals and their challenges.

Author Response

Question 1: Moderate English changes required
Answer 1: The English of the manuscript has been improved by an
English native speaker. The manuscript originates in a common effort
shared by basic and clinical scientists and intends to highlight
synergizing evidence from distant fields, as requested by the editors of
the Special Issue.
Question 2: This review is a very difficult read. Each section is a
collection of observations/statements and the authors quickly switch
from one idea to the next. There seems to be lot of information but this
needs to be organized to convey one main idea per section. At the
moment there is lack of flow and clarity. Find the main idea that you
wish to convey in each section and summarize findings from the
literature with a good structure and flow between the statements. The
only exception is section 5 concerning flow cytometry which reads
coherently and is well structured.
Answer 2: The presentation is organized chronologically, highlighting

biological evidence potentially transferrable to clinics. Flow cytometry
is a promising tool for clinical applications for the analysis of other
still open issues.
Question 3: Several sentences lack proper references. For example,
page 2 the paragraph starts with “Study of function…” and “Another
approach…”. Similarly, page 6 “It has also been…”. These paragraphs
do not have references. There are similar instances throughout the
review.
Answer 3: The text has been amply re-edited and organized, taking
into close consideration reviewer’s statements. Appropriate references
are now included in the text and in the reference list.
Question 4: Another problem is that the authors cite reviews and when
one goes back to the review that article lacks a proper reference as
well. As an example in page 3 “One of the first molecules…” cites
reference 20. Reference 20 simply states this fact without showing data
or citing a reference. Cite the article that describes the original work
showing expression of these two molecules. Check references
thoroughly to insure original work is cited.
Answer 4: The reference list has been modified according to reviewer
suggestion.
Question 5:What is 39 in Fig 1? There is no reference relating to
Atanackovic D in the list of references
Answer 5: the reference list has been modified and all references are
now cited in the appropriate section of the manuscript.
Question 6: The authors have self-cited more than 20 times, which is
excessive. Please be careful when self-citing.
Answer 6: Self citation has been reduced.
Question 7: Provide some explanation regarding CDC, ADCC.
Answer 7: ADCC and CDC have been explained
Question 8: Page 4 “These contradictory findings…” the scorpion
effect is not clear. Explain it clearly
Answer 8: The Scorpion effect has been eliminated
Question 9: The authors provide a rather extensive history of CD38.
This could be shortened and the space can be utilized to provide
summary of the ongoing trials with CD38 monoclonals and their
challenges.
Answer 9: The section on the history of CD38 history has been
reduced, as have the self-citations. The other references have been
checked and modified accordingly.
Ongoing clinical trials are now mentioned, and the clinical part of this
manuscript has been expanded.

I do hope that the revised version of the manuscript might match your
approval

Round 2

Reviewer 3 Report

Dear authors,

The content has been appropriately modified and it reads much better than the previous version.

In fig1 legend, please include conclusions in text and explain only the figure in the legend. Also the term autovaccination needs to be explained and/or referenced in the text before introducing it in the figure.

Author Response

Question:In fig1 legend, please include conclusions in text and explain only the figure in the legend. Also the term autovaccination needs to be explained and/or referenced in the text before introducing it in the figure.

Answer:Legend to Figure 1 has been modified, accordingly, An explanation of the term "auto vaccination" has been added in the text.